# Pharmacokinetics of Haloperidol in Critically Ill Patients: Is There an Association with Inflammation?

**DOI:** 10.3390/pharmaceutics14030549

**Published:** 2022-02-28

**Authors:** Letao Li, Sebastiaan D. T. Sassen, Mathieu van der Jagt, Henrik Endeman, Birgit C. P. Koch, Nicole G. M. Hunfeld

**Affiliations:** 1Department of Hospital Pharmacy, Erasmus MC-University Medical Center, Doctor Molewaterplein 40, 3015 GD Rotterdam, The Netherlands; l.li.1@erasmusmc.nl (L.L.); s.sassen@erasmusmc.nl (S.D.T.S.); b.koch@erasmusmc.nl (B.C.P.K.); 2Department of Intensive Care, Erasmus MC-University Medical Center, Doctor Molewaterplein 40, 3015 GD Rotterdam, The Netherlands; m.vanderjagt@erasmusmc.nl (M.v.d.J.); h.endeman@erasmusmc.nl (H.E.)

**Keywords:** haloperidol, pharmacokinetics, delirium, critical ill, ICU

## Abstract

Haloperidol is considered the first-line treatment for delirium in critically ill patients. However, clinical evidence of efficacy is lacking and no pharmacokinetic studies have been performed in intensive care unit (ICU) patients. The aim of this study was to establish a pharmacokinetic model to describe the PK in this population to improve insight into dosing. One hundred and thirty-nine samples from 22 patients were collected in a single-center study in adults with ICU delirium who were treated with low-dose intravenous haloperidol (3–6 mg per day). We conducted a population pharmacokinetic analysis using Nonlinear Mixed Effects Modelling (NONMEM). A one-compartment model best described the data. The mean population estimates were 51.7 L/h (IIV 42.1%) for clearance and 1490 L for the volume of distribution. The calculated half-life was around 22 h (12.3–29.73 h) for an average patient. A negative correlation between C-Reactive Protein (CRP) and haloperidol clearance was observed, where clearance decreased significantly with increasing CRP up to a CRP concentration of 100 mg/L. This is the first step towards haloperidol precision dosing in ICU patients and our results indicate a possible role of inflammation.

## 1. Introduction

Delirium is quite common in intensive care unit (ICU) patients and is associated with poor clinical prognosis [1,2,3,4,5]. Currently the treatment of delirium may include pharmacological agents, including antipsychotics, melatonin, alpha-2 agonists (dexmedetomidine and clonidine), next to nonpharmacological interventions [6,7,8]. Among antipsychotics, haloperidol is the most commonly used. However, clinical evidence for the effect of haloperidol in decreasing ICU delirium is scarce [8,9,10,11]. Pharmacokinetics (PK) can play an important role in understanding the effect of haloperidol in ICU patients. Critically ill patients tend to show large differences in PK [12,13]. In the case of haloperidol, this may lead to increased variability in haloperidol blood concentrations in ICU patients, compared to non-ICU patients [14,15]. The variable PK might explain the variability in the effect, hence adjusting the dose based on individual PK parameters might improve drug efficacy. To understand more of this variability in blood concentrations, it is important to specifically study the pharmacokinetics of haloperidol in ICU patients.

Previous studies in non-ICU populations have shown that haloperidol has typical pharmacokinetic features of a lipophilic drug. It has high protein binding (90%), large volume of distribution (Vd) (1000–3000 L), and is predominantly metabolized by the liver and gut via glucuronidation (40–50%), CYP3A4 (25–30%), and CYP2D6 (25–30%) [15,16,17,18]. Previous studies have shown that CYP2D6 genetic polymorphism influences the haloperidol concentration levels in non-ICU patients [19,20]. For CYP3A4, the isoenzyme activity caused concentration changes only at higher doses [21] The glucuronidation is a major metabolism pathway of haloperidol [22], but only in vitro studies have shown that this pathway might cause inter-individual concentration variance [23]. The change in volume of distribution caused by pathophysiological changes is relatively small compared to hydrophilic drugs, but it is more susceptible to changes in drug clearance (CL) due to liver function alteration.

The available pharmacokinetic parameters related to haloperidol are mainly from studies in a healthy population or in non-critically ill patients with schizophrenia with relatively small sample sizes (less than 10 patients). Only one study on Japanese psychopaths included 218 patients [16,24,25,26,27,28,29]. Hence, the dosing strategy based on the parameters from those models might not be suitable for ICU patients. The goal of this study is to better understand the pharmacokinetics of haloperidol in critical care patients. Information on pharmacokinetics by means of population PK modelling may support further efficacy studies of haloperidol in critically ill patients.

## 2. Materials and Methods

### 2.1. Study Design

Data were collected at the adult ICU of Erasmus University Medical Centre (EMC), Rotterdam, the Netherlands, during a 3-year period (between October 2014 and April 2017) as previously described [30]. This study was conducted in accordance with the principles of the Declaration of Helsinki (version: October 2008) and approved by the Institutional Review Board (project identification code: MEC-2014-264, 21-Juli-2014, Medisch Ethische Toetsings Commissie Erasmus MC), more details see Appendix A. Informed consent was obtained from each patients’ legally authorized representative given all patients had delirium (see Appendix A). Patients who developed delirium received 1 mg every 8 h (q8h) by intravenous bolus infusion [or 0.5 mg q8h by intravenous bolus infusion for patients aged ≥ 80 years or 2 mg q8h by intravenous bolus infusion in case of agitation] within 8 h of delirium detection, which constituted the routine regimen in the EMC for the treatment of ICU delirium at that time. The haloperidol dose was decreased if the Intensive Care Delirium Screening Checklist (ICDSC) score, a validated screening tool for delirium, was below or equal to 3 for more than 24 h, and was ceased if the ICDSC was below or equal to 3 for more than another 24 h.

### 2.2. Data Collection

Samples were collected and determined on days 2, 3, 4, 5, and 6 (end of study) each morning before haloperidol was dosed or discontinued according to protocol standards, or in participants who were discharged from ICU or transferred to another hospital.. The pharmacokinetic modeling was performed at day 2 (t = 0–1 h, t = 2–3 h, t = 4–5 h, t = 6–8 h). Serum samples were collected in EDTA tubes. Samples were collected from the arterial line in a strictly standard manner (according to ICU protocol) by ICU nursing staff and therefore we expect no infection risk. The samples were immediately sent to the laboratory of the hospital pharmacy and stored at −80 °C and then thawed at room temperature before analyzing. Levels of albumin, creatinine, urea, bilirubin, and C-reactive protein (CRP) where measured in LiHep plasma (Barricor Vacutainer, BD, Franklin Lakes, NY, USA, Belgium) on a routine chemistry analyzer (Cobas 8000, Roche Diagnostics, Basel, Switzerland). Additionally, white blood cell-(WBC) and platelet count were analyzed according to standard clinical care in the ICU in whole blood (K2EDTA Vacutainer, Franklin Lakes, NY, USA) on a routine hematology analyzer (XN9000, Sysmex, Kobe, Japan).

Serum haloperidol concentrations were analyzed via validated Liquid Chromatography tandem Mass Spectrometry (Waters Corporation, Milford, MA, USA) (see Appendix A). We determined linearity, lower limit of quantitation (LLOQ), upper limit of quantitation (ULOQ). The method was validated according to Food and Drug Administration(FDA)/European Medicines Agency (EMA) guidelines [31,32]. The 2.1 × 100 mm Waters Acquity UPLC BEH C18 1.7 μm column (cat no. 186002352) was used in combination with optimized chromatographic conditions. To suit the validation parameters for analytical validation, a shorter runtime of 5 min and the use of two eluents with changing percentage was tested. In addition, we optimized the method for a higher sensitivity and selectivity, according to the standard procedure for validation of our method. The LLOQ was set at 0.5 µg/L and the ULOQ at 20 µg/L. Other parameters that were collected were age, gender, ethnic origin, Body Mass Index (BMI), Acute Physiology and Chronic Health Evaluation (APACHE) III score, Intensive Care Delirium Screening Checklist (ICDSC) score at start of haloperidol, quetiapine exposure, additional drug use, CYP450 status, admission reason, length of ICU stay, amount of blood samples collected, and outcome.

### 2.3. Data Analysis

The pharmacokinetic analysis of haloperidol was performed using the nonlinear effects modeling approach in NONMEM^®^ first-order conditional estimates (FOCE) with interaction [version 7.4, ICON, Development Solutions, MD, USA], Pirana version 2.9.9 (Certara, Princeton, NJ, USA), and data were further analyzed in R version 4.0.5 (R Foundation for Statistical Computing, Vienna, Austria). All the concentration data were log-transformed. A one-compartment model was fitted to the data. Subsequently, more complex models were tested. The model fit was evaluated both numerically by the precision of the estimated PK parameters and the change in the objective function values (dOFV), and visually by goodness-of-fit plots (GoF) and visual predictive checks (VPC). For the covariate analysis, the stepwise covariate modeling with forward inclusion-backward elimination method was applied [33]. In the forward process, a 3.84-point decrease in OFV for one degree of freedom was considered a significant improvement of the model with a *p*-value of <0.05. For the backward elimination process, the statistical criterion was set to an increase of OFV to 6.64 for covariate selection. A constant error model was used on the log transformed data to describe the residual error in the model predicted plasma concentrations. Age, gender, length, weight, BMI, BSA, CYP3A4, CYP2D6, CPR, creatinine, albumin, bilirubin, APACHE III, SOFA, ASAT, ALAT, potential interaction drugs (including erythromycin, amiodarone, metoprolol, metoclopramide, voriconazole, and fluconazole), WBC, and platelet count were tested as covariates. CYP3A4 and CYP2D6 patient genotyping was performed using Autogenomics INFINITY genotyping platform (Carlsbad, CA, USA) and relevant alleles present and gene duplication were detected. Patients were classified according to the number of active enzyme alleles present: poor metabolizers (PM; two defective alleles), intermediate metabolizers (IM, 2 decreased activity alleles or 1 active and 1 inactive allele), extensive metabolizers (EM), and ultra-rapid metabolizers (UM, gene duplication positive in the absence of a CYP2D6 null allele).

### 2.4. Model Simulation

To show an illustration of the covariate effect on the plasma concentrations of haloperidol, deterministic simulations were performed by using NONMEM. The haloperidol plasma concentrations under different covariates were simulated over a time course of 72 h (last dose at 72 h) and intravenous bolus doses were administered every 8 h. The median and 90% confidence interval are shown graphically.

## 3. Results

### 3.1. Study Population

An overview of all patient characteristics is presented in Table 1. A total of 22 critically ill adult patients were enrolled in the study; 54.5% of the patients were male, median age was 67 years (range from 48 to 77), median BMI was 27 (range from 18 to 39) m^2^, median APACHE III score was 80.5 (range from 54 to 181), median length of ICU stay was 16 days (range from 2 to 63). Main reasons for ICU admission were surgery (*n* = 7; 32%), sepsis (*n* = 3; 14%), respiratory failure (*n* = 3; 14%), and vascular aneurysm (*n* = 2; 9%). For the CYP2D6 status: extensive metabolizers (*n* = 12, 54%), intermediate metabolizers (*n* = 7, 32%), and poor metabolizers (PM) (*n* = 3, 14%). No ultra-rapid metabolizers were detected. For the CYP3A4 status: extensive metabolizers (*n* = 18, 82%) and intermediate metabolizers (*n* = 4, 18%). No ultra-rapid metabolizers or PMs were detected. Of the 22 patients, eleven patients died (50%), of which six during the ICU stay, four after ICU discharge and one after transfer to another hospital.

Total daily intravenous doses of haloperidol ranged from 1.5 to 6 mg. A total of 145 blood samples were collected and a total of 6 concentrations were censored due to sampling errors (sampling during the haloperidol infusion).

### 3.2. Structural Model

The logarithmic transformed concentration data were best described by a one-compartment model with an additive residual error. Inter-individual variability (IIV) was included on CL. The final structural model was used for covariate analysis. Stepwise (forward and backward screening) model building strategies were implemented to identify potential covariates, explaining the between-subject variability in model parameters, equations, and model codes (Appendix A, respectively).

The potential covariates (age, gender, length, weight, BMI, BSA, CYP3A4, CYP2D6, CPR, creatinine, albumin, bilirubin, APACHE III, SOFA, ASAT, ALAT, potential interaction drugs (including erythromycin, amiodarone, metoprolol, metoclopramide, voriconazole and fluconazole), WBC, and platelet) were screened. For continuous variables such as WBC, CRP, and weight, we used the value divided by the median as covariate on the clearance. For categorical variables like gender, CYP3A4, and CYP2D6 polymorphism, we gave different variables a value and multiplied it with typical haloperidol clearance value. Only CRP on CL resulted in a significant improvement of model fit, with a drop in OFV of 7.533 and a decrease in IIV on CL from 40.4% to 29.9%. Adding CRP into the equation decreased the objective function value (OFV) from −49.32 to −56.81 (dOFV = −7.49), which explained 31% of the IIV on CL. An overview of all parameter estimates is given in Table 2. The clearance of haloperidol is 51.7 L/h and it has a large volume of distribution (1490 L). CRP was able to significantly decrease the variance in CL, as is shown in Figure 1. When CRP was incorporated as a covariate into the final model, the ETA decreased and became more evenly distributed.

### 3.3. Model Evaluation

Figure 2 shows that both the population predictions (PRED) and the individual predictions (IPRED) were evenly distributed around the uniform line when plotted versus observed concentrations (DV). The weighted residuals were symmetrically distributed throughout the time after dose and prediction errors were predominantly within two standard deviations. Figure 3 shows the covariate CRP on the effect of haloperidol clearance. There is a negative relationship between the clearance and CRP; the relationship disappeared with CRP above 100 mg/L.

The results of the bootstrap (*n* = 1000) were in accordance with the estimates of the original model data. A visual predictive check (VPC) was executed (Figure 4) to validate the model by simulating 1000 data sets, comparing the observed concentration with the distribution of simulated concentrations [34]. Figure 4 shows the VPC results and the model fitted well.

### 3.4. Simulations

The simulation results of the concentration under 1 mg of intravenous administered haloperidol every 8 h are shown in Figure 5. In Figure 5a, the concentration of the haloperidol increased from 1 ng/L to 2 ng/L when the CRP increased from 5 mg/L to 100 mg/L. Figure 5b shows two different simulation patients—in the left graph, a change in CRP from 5 mg/L to 100 mg/L during unaltered haloperidol dosing; the right is the opposite with CRP changing from 100 mg/L to 5 mg/L. The latter requires a longer time to reach a steady state of haloperidol concentration.

## 4. Discussion

This is the first study describing the pharmacokinetics of low dose haloperidol in adult critically ill patients. A one-compartment model adequately described the pharmacokinetics of haloperidol with good accuracy. The most interesting finding was the negative correlation between clearance and CRP levels (as long as it remained below 100 mg/L). This may indicate that a low level of inflammation may play a role in the pharmacokinetics of haloperidol.

The PK parameters of haloperidol from our final model were CL (51.7 L/h), Vd (1490 L), and t1/2 (22 h), which is similar to a previous report [29]. The negative effect of increased CRP on CL reached its maximum at around 50 mg/L to 100 mg/L after which CL did not decrease with further increasing CRP concentrations. It has been proven that inflammation influences the PK of many drugs (midazolam, irinotecan, clozapine, quetiapine, risperidone, voriconazole, perampanel) by changing the distribution of volume, influencing the enzyme activity and hepatic/renal blood flow and thus influencing the drug metabolism and excretion [35,36,37,38,39,40,41,42]. The phenomenon of CRPs negative relationship with haloperidol concentration in our study implies that inflammation can influence the clearance of haloperidol in ways other than liver function, as no significant correlation was found between clearance and liver function indicators (ASAT, ALAT, serum bilirubin), which is similar to the results of L.G. Franken et al. (28). However, other inflammatory markers such as leukocytes and platelets did not show any correlation in our study. The clearance did not further decrease with increased CRP, which is probably because the inflammation effect on clearance had reached its max effects. So far, we have no clear explanation for this phenomenon and unfortunately, there are no other data available on this topic.

The CRP could help us better estimate drug exposure and lead to more precise individual dosing. In lower levels of inflammation, generally indicating less sick patients, relatively lower CRP levels might require higher drug dosing versus higher CRP levels, given that lower CRP results in lower trough levels. This is indeed a clinically relevant signal, but requiring confirmation and external validation. Another important issue is that the haloperidol therapeutic target concentration remains unknown with respect to delirium and requires additional research as well.

Besides the possible association between CRP and clearance, the pharmacokinetics parameters (CL = 51.7 L/h, V = 1490 L) of our study are similar to the results of previously published haloperidol models (CL range from 42.4 L/h to 88 L/h, V range from 2060 L to 3169 L) in a healthy population and studies on schizophrenia [29,43]. Furthermore, we found no correlation between clearance and other factors, such as co-medication or different CYP genotypes. However, other covariates, such as bodyweight, which were shown to be important in other studies [43,44], did not show significant associations with clearance in our study. This is most likely due to the limited number of patients, the low dose of haloperidol, and limited samples, in combination with the heterogeneous population in the ICU. On the other hand, the parameters in our study differ considerably from the parameters (CL = 29.3 L/h, V = 1260 L) of studies on terminally ill patients [28]. This difference might be explained by the impaired (reduced) liver function (liver capacity) of terminally ill patients, resulting in a decreased haloperidol clearance.

One limitation of our study was the limited number of patients, which might explain why some potential important covariates did not show significance in our model. Furthermore, the published haloperidol population models all use the two-compartment model; however, in this study, owing to the small dataset, we were unable to accurately describe a peripheral compartment and inter-compartmental clearance. However, the one-compartment model fit the data well. Furthermore, we did not look at the pharmacodynamic effect so we could not link the concentration to the haloperidol toxicity and delirium symptom relief, since the therapeutic target of haloperidol is unknown. In addition, we only detected the whole blood concentration, not the free fraction of the haloperidol. Future research should also take this into account.

It is necessary to find more accurate delirium severity related markers or clinical scores which could explore the haloperidol concentration and its effect/response relationship and whether the pharmacokinetic data could be extrapolated to higher doses/concentration range. Further studies on pharmacokinetics and pharmacodynamics of higher-dosed haloperidol in ICU patients with delirium are warranted in order to more accurately assess efficacy.

## 5. Conclusions

This study describes the pharmacokinetics of low dose haloperidol in critically ill patients with adequate accuracy and showed that clearance is negatively related to CRP at low levels (0–100 mg/L), which seems to indicate a role of inflammation on haloperidol pharmacokinetics.

## Figures and Tables

**Figure 1 pharmaceutics-14-00549-f001:**
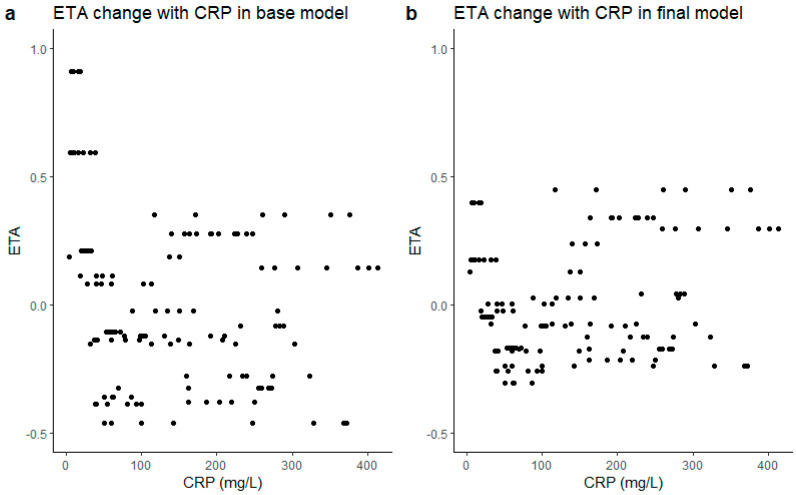
ETA of CL versus CRP: (**a**) ETA versus CRP in the base model; (**b**) ETA versus CRP in the final model. CRP: C−reactive protein.

**Figure 2 pharmaceutics-14-00549-f002:**
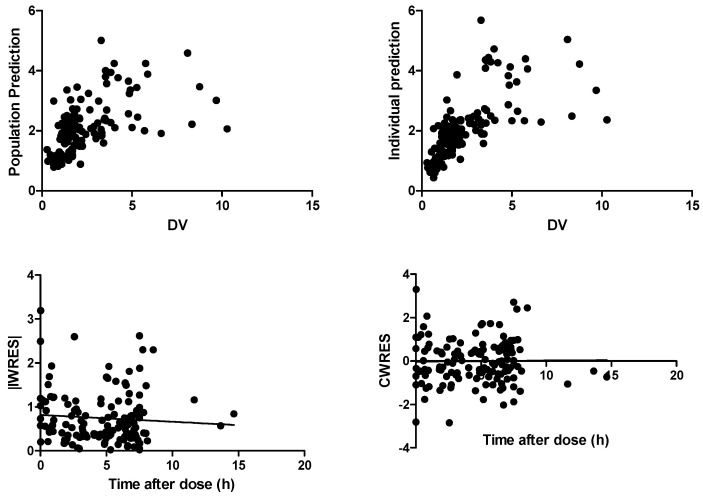
Basic goodness of fit plots for the final model: population predictive concentration versus observed concentration (DV) (**upper left**); individual predictive concentration versus observed concentration (DV) (**upper right**); time after dose versus individual weighted residuals (IWRES) (**lower left**); time after dose versus conditional weighted residuals (CWRES) (**lower right**). CWRES: conditional weighted residuals, DV: dependent variable, IWRES: individual weighted residuals.

**Figure 3 pharmaceutics-14-00549-f003:**
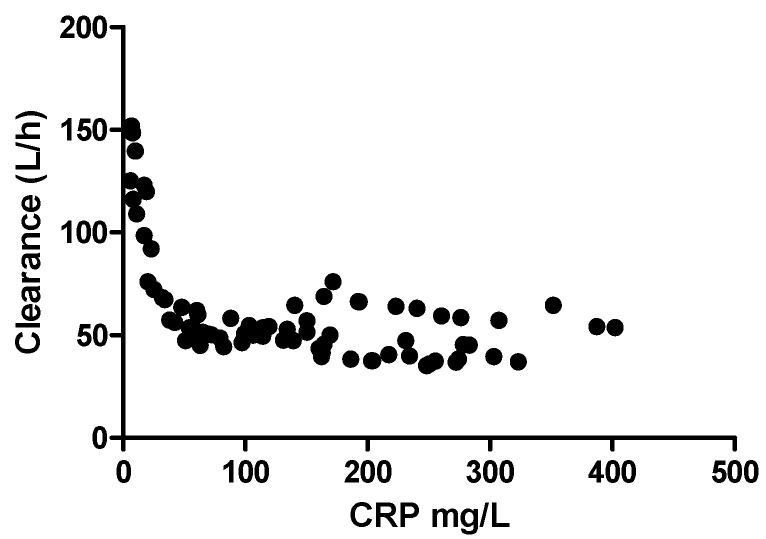
The relationship of inflammatory indicator CRP and haloperidol clearance. CRP: C-reactive protein.

**Figure 4 pharmaceutics-14-00549-f004:**
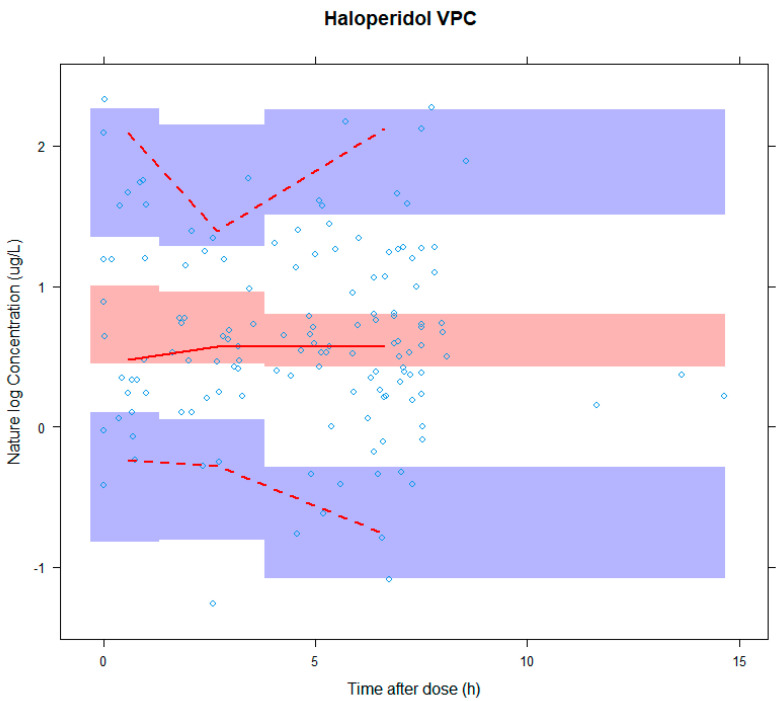
The visual predictive check of haloperidol. The x−axis is time (h) and y−axis is concentration of haloperidol in natural log scale. VPC: visual predictive check.

**Figure 5 pharmaceutics-14-00549-f005:**
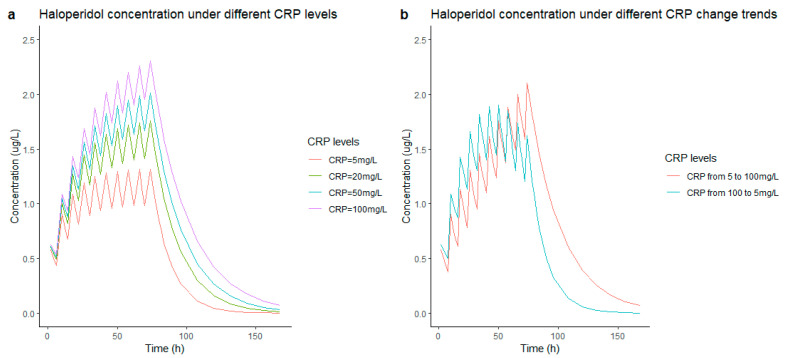
The simulation of the influence of CRP levels on haloperidol concentration. (**a**) is to use the final model to simulate concentration with different CRP levels (5, 20, 50, 100 mg/L), (**b**) shows two different simulations—on the left, an increase in CRP from 5 mg/L to 100 mg/L while on the right an increase in CRP from 100 mg/L to 5 mg/L; the increasing/decreasing rate is 20 mg/L per 12 h. All haloperidol simulations are performed at a dose of 1 mg q8h up until 72 h, the median concentrations are used to plot the simulation. CRP: C-reactive protein.

**Table 1 pharmaceutics-14-00549-t001:** Patient characteristics over the time course of the study.

Characteristics	N = 22
Age, years (median, range)	67 (48–77)
Male, *n*(%)	12 (54.5)
Female, *n*(%)	10 (45.5)
Weight, kg (median, range)	80 (52–137)
Ethnic origin, *n* (%)	
Caucasian	22 (100)
BMI (median, range)	27 (18–39)
Primary reason for ICU admission, *n* (%)	
Surgery	7 (32%)
Respiratory failure	3 (14%)
Sepsis	3 (14%)
Vascular aneurysm	2 (9%)
Blood chemistry, serum levels at admission (median, range)	
Albumin, g/L	26 (6–47)
Creatinine, μmol/L	130 (32–401)
Urea, mmol/L	13(4–46)
Bilirubin, μmol/L	14 (3–754)
CRP, mg/L	171 (4.1–368)
CYP2D6 *n* (%)	
Extensive metabolizers	12 (54%)
Intermediate metabolizers	7 (32%)
Poor metabolizers	3 (14%)
CYP3A4 *n* (%)	
Extensive metabolizers	18 (82%)
Intermediate metabolizers	4 (12%)
Quetiapine exposure *n*(%)	5 (22.7%)
APACHE III score median (range)	81 (76–99)
ICDSC baseline median (range)	4 (1–6)
Duration of stay (during using halo), days (median, range)	6.5 (3–8)
Died in ICU, *n* (%)	11 (50)
Cause of death	
Respiratory failure (During ICU)	1
Sepsis with multiple organ failure (During ICU)	4
Cardiac causes (after ICU)	2
Gastrointestinal causes (after ICU)	2
Respiratory insufficiency (transferred to another hospital)	1
Unknown (transferred to another hospital)	1
Blood samples collected, median (range)	7.5 (3–8)

APACHE: Acute Physiology and Chronic Health Evaluation, BMI: Body mass index, CRP: C-reactive protein, ICDSC: Intensive Care Delirium Screening Checklist, ICU: intensive care unit.

**Table 2 pharmaceutics-14-00549-t002:** Population pharmacokinetic parameters for base and final models.

Parameter	Base Model	RSE%	Shrinkage%	Final Model	RSE%	Shrinkage%	Bootstrap of the Final Model		
							Median	90% Percentile (Lower)	90% Percentile (Upper)
CL (L/h)	54.6	11		51.7	12		50.64	39.65	63.74
Vd (L)	1450	29		1490	31		1522.05	893.6	2305.2
CRP				−0.23	50		−0.21	−0.02	−0.42
IIV-CL (%)	40.4%	31	15	29.9%	27	24			
Residual variability	0.457	9	6	0.461	9	5	0.446	0.382	0.54

CRP: C-reactive protein, CL: clearance, IIV-CL: inter-variability on clearance, Vd: volume of distribution.

## Data Availability

The data that support the findings of this study are available from the corresponding author upon reasonable request.

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
