# Peer review of "Pharmacokinetics of Haloperidol in Critically Ill Patients: Is There an Association with Inflammation?"

_pharmaceutics, 2022, doi:10.3390/pharmaceutics14030549_

Round 1

Reviewer 1 Report

This is the first study describing the pharmacokinetics of haloperidol in adult critically ill patients. The authors should complete the following data:

  1. In the introduction, the PK data of haloperidol should be completed and more data on drug interactions by isoenzymes and transporters should be provided.
  2. I do not have the attachments in the supplement that are provided in the article. Data must be provided: reagents, sample treatment method, validation conditions and results, chromatograms, graph. 3. What are the limitations of the study? 4. Why were the free fraction of the drug and haloperidol metabolites not determined?

Author Response

Dear Reviewer,

thanks for your comments and suggestions, Please see the attachment

Best wishes,

Reviewer 2 Report

This study described the haloperidol PK model that has the potential to contribute to the safe and effective haloperidol dosing strategy for ICU patients. The comments were added as below.

Abbreviation: please spell out “WBC (line 78)”, “LLOQ (line 80)”, “FDA (line 81)”, “BMI (line 82)”, “APACHE (line 82), etc according to authors guidelines.

Line 58: “intensive care units” can be abbreviated.

Line 64: “1 mg IV bolus q8h” should be presented as “ I mg every 8 hours (q8h) by intravenous bolus infusion”

Line 101. How did the authors use “CYP3A4” and “CYP2D6” as covariates?

Table 1. Please clarify the cause of death for “Died in ICU” in the table.

Based on the previous reports (https://pubmed.ncbi.nlm.nih.gov/24113674/, https://link.springer.com/article/10.1007/s00228-017-2283-6, etc), please discuss the difference or consistent facts (CL and Vd values) with this current report in one paragraph.

The author should evaluate the QT interval pharmacodynamics after haloperidol infusion as a safety assessment. Furthermore, discuss whether the new regimen can improve the patient’s outcome.

Can the new regimen considering the CRP statement improve the patient’s outcome? Can this study result contribute to the medication strategy for delirium?

Author Response

Dear Reviewer,

Thanks for your comments, Please see the attachment.

Best wishes

Round 2

Reviewer 2 Report

How did the authors determine the CYP status?

Please describe the CYP status of the patient's background. I did not find this information.

Almost all revisions were addressed.

Author Response

Dear Reviewer,

For your question, Please see the attachment
